# Impact of a perioperative oral opioid substitution protocol during the nationwide intravenous opioid shortage: A single center, interrupted time series with segmented regression analysis

Reza Salajegheh[1], Edward C. Nemergut[1,2], Terran M. Rice[3], Roy Joseph[3,4], Siny Tsang[5], Bethany M. Sarosiek[6], C. Paige Muthusubramanian[6], Katelyn M. Hipwell[3], Kate B. Horton[3], Bhiken I. Naik[1,2]*

1 Department of Anesthesiology, University of Virginia, Charlottesville, Virginia, United States of America, 2 Department of Neurological Surgery, University of Virginia, Charlottesville, Virginia, United States of America, 3 Department of Pharmacy, University of Virginia, Charlottesville, Virginia, United States of America, 4 Department of Pharmacy, Rady Children's Hospital–San Diego, San Diego, California, United States of America, 5 Department of Nutrition and Exercise Physiology, Washington State University, Spokane, Washington, United States of America, 6 Department of Perioperative Services, University of Virginia, Charlottesville, Virginia, United States of America

* bin4n@virginia.edu

## Abstract

### Introduction

To mitigate the recent nationwide shortage of intravenous opioids, we developed a standardized perioperative oral opioid guideline anchored with appropriate use of nonopioid analgesia, neuraxial and loco-regional techniques. We hypothesize that adoption of this new guideline was associated with: 1) equivalent patient reported pain scores in the post-anesthesia care unit (PACU); and 2) equivalent total opioid use (oral and parenteral) during the perioperative period.

### Methods

Cases performed from July 1, 2017 to May 31, 2019 were screened. All opioids administered were converted to intravenous morphine milligram equivalents. Segmented regression analyses of interrupted time series were performed examining the change in opioid use, PACU pain scores and number of non-opioid analgesic medications used before and after the protocol implementation in April 2018.

### Results

After exclusions, 29, 621 cases were included in the analysis. No significant differences in demographic, ASA status, case length and surgical procedure type were present in the pre and post-intervention period. A significant decrease in total (Estimate: -39.9 mg, SE: 6.9 mg, p < 0.001) and parenteral (Estimate: -51.6 mg, SE: 7.1 mg, p < 0.001) opioid use with a

**Data Availability Statement:** All relevant data are within the manuscript and its Supporting Information files.

**Funding:** The author(s) received no specific funding for this work.

**Competing interests:** The authors have declared that no competing interests exist.

significant increase in oral opioid use (Estimate: 9.4 mg, SE: 1.1 mg, p < 0.001) was noted after the intervention. Pain scores were not significantly different between the pre- and post-intervention period (Estimate: 0.05, SE: 0.13, p = 0.69).

## Conclusion

We report our experience with a primary perioperative oral based opioid regimen that is associated with decreased total opioid consumption and equivalent patient reported pain scores.

## Introduction

The recent shortage of intravenous opioids triggered by unanticipated manufacturing delays and increasingly restrictive federal drug policy resulted in anesthesia providers managing patients with critically low supplies of intravenous opioids. [1, 2] Alternate approaches to managing this parenteral opioid shortage included increased utilization of oral opioids, use of transdermal fentanyl and addition of nonopioid adjuncts to hospital formularies. [1, 3]

In contrast to the impact of an oral opioid substitution protocol in a nonsurgical cohort, there is limited data on the impact of limited parenteral opioids on pain management during and after a surgical intervention. [4] We present our data over a 23-month period (9 months pre-intervention and 14 months post-intervention) following institution of a guideline developed by our Acute Pain Service and the Inpatient Pharmacy in response to the acute, limited supply of parenteral opioids. The goals of the protocol were to create an intravenous to oral opioid substitution guideline for intraoperative and immediate postoperative pain management, facilitate increase use of nonopioid analgesia and ensure that neuraxial and loco-regional techniques were utilized when appropriate. We hypothesize that adoption of this new guideline was associated with: 1) equivalent patient reported pain scores in the post anesthesia care unit (PACU); and 2) equivalent total opioid use (oral and parenteral) during the perioperative period.

## Methods

### Study design

We performed a single center, interrupted time series with segmented regression analysis investigating opioid, nonopioid, loco-regional and neuraxial utilization at the University of Virginia prior to and following implementation of protocol designed to limit intravenous opioid use.

**Ethics statement.** The institutional review board at the University of Virginia waived the requirement for written informed consent and approved the study. The study approval number is HSR #21976. Data was not anonymized before accessing it. The source of the medical records analyzed in this work were obtained only from the University of Virginia Health Center. The STROBE statement checklist for reporting observational studies was followed throughout this study and reported in the S1 Checklist.

### Study population

All cases documented in our electronic medical record from July 1, 2017 to May 31, 2019 were initially screened. The new guideline was implemented on April 1, 2018. We excluded cases

that were performed on patients < 18 years, American Society of Anesthesiologist (ASA) Physical Status 5 and 6 patients, those without recorded PACU pain scores within 2 hours after surgery, patients undergoing cardiac surgery and those who remained intubated postoperatively.

## Study intervention

The Limited Intravenous Opioid Guideline, described below, was developed in collaboration with the Acute Pain Service and the Pharmacy Department. Prior to implementation of the protocol, a meeting with all stakeholders (anesthesia providers, surgeons, pharmacy, preoperative and postoperative nursing staff) were held and active feedback was solicited. The final protocol was approved by all stakeholders prior to the implementation date.

**Limited intravenous opioid guideline.** The guideline was anchored by the following principles: 1) Increase use of multimodal, nonopioid adjuvants prior to surgery, during the procedure and in the PACU, 2) Use of short and long acting enteral and rectal opioids prior to surgery, during the procedure and in the PACU, 3) The use of loco-regional and neuraxial techniques, when appropriate. These components were implemented through the three major perioperative phases of care:: pre-operatively, intraoperatively, and in PACU. The choice and combination of analgesic agents utilized were left at the discretion of the anesthesia providers.

## Preoperative phase of care (surgical admissions suite)

**1. Non-opioids.** Unless contraindicated, all patients with an anticipated postoperative inpatient admission and with a surgical duration greater than 2 hours received the following nonopioid analgesics orally: gabapentin 600 mg, celecoxib 200 mg and acetaminophen 975 mg. If the expected surgical duration was less than 2 hours or patients were planned to be discharged after the procedure, they received the aforementioned combination of nonopioid analgesics however gabapentin was reduced to 300 mg.

**2. Opioids.** Oral opioids were administered prior the procedure based on the expected duration of surgery and the postoperative destination status (in-patient requiring admission vs. outpatient being discharged on the day of procedure)

## Procedures less than two hours and/or outpatient status

a. Oxycodone 5 mg-10 mg or hydromorphone 2–4 mg orally.

## Procedures greater than two hours and/or inpatient status

a. Methadone 5–10 mg orally or

b. Extended release morphine 15–30 mg orally or

c. Extended release oxycodone 10–20 mg orally

d. Patients undergoing cardiac or complex spine surgery received higher doses of oral methadone (0.2 to 0.3 mg/kg).

Intraoperative Phase of Care:

1. Use of esmolol bolus (10–50 mg) to blunt intubation response with avoidance of intravenous opioids

2. Intravenous ketamine infusions 0.1–0.3 mg/kg/hour, if not contraindicated

3. Intravenous lidocaine infusions ranging from 1–3.5 mg/min, if not contraindicated

4. Intravenous dexmedetomidine 0.3–0.7 mcg/kg/h, if not contraindicated.

5. The use of nonsteroidal anti-inflammatory agents (ketorolac 15–30 mg) after discussion with the surgical team and if no celecoxib was administered preoperatively.

6. Use of morphine suppository (10–20 mg), 30–60 minutes prior to the end of the case if patient demonstrated features of inadequate analgesia such as persistent hypertension and tachycardia.

7. The use of loco-regional techniques when appropriate at the end of cases (e.g. transversus abdominus plane and rectus sheath blocks)

8. Intravenous fentanyl or hydromorphone, after approval from the attending anesthesiologist, if aforementioned interventions contra-indicated or not effective in providing analgesia (persistent hypertension and tachycardia)

PACU Phase of Care:

1. For acute postoperative pain: oral oxycodone 5–10 mg or oral hydromorphone 2–4 mg with or without acetaminophen 975 mg (if timing appropriate from preoperative phase of care dose). Where appropriate, repeated dose of nonsteroidal anti-inflammatory agents was encouraged.

2. At the discretion of the anesthesia provider, continuation of the lidocaine infusions at 0.5–1 mg/min or ketamine infusions at 0.1 mg/kg/hour.

3. For acute severe postoperative pain: administration of intravenous fentanyl or hydromorphone, after approval from the attending anesthesiologist. Addition of ketamine 20 mg intravenous once (with accompanying midazolam as needed).

## Data variables

Demographic data collected included age, gender, height and weight. ASA physical status, surgical case type and case length (start of case to end of case) were recorded. ASA physical status, case length and surgical case type were used as a surrogate for case complexity.

Non-opioid medications including acetaminophen, celecoxib, dexmedetomidine, esmolol, gabapentin, ketamine and lidocaine administered preoperatively, intraoperatively and postoperatively in the PACU were recorded. Data for any loco-regional or neuraxial procedures performed either pre, intra or postoperatively in the PACU were collected.

## Study outcomes

All pre, intra and postoperative opioids including fentanyl, sufentanil, alfentanil, morphine, hydromorphone, meperidine, remifentanil, methadone, tramadol and oxycodone were recorded. Opioid doses administered with either epidural or spinal anesthesia prior to the procedure were also included. All opioids administered via the oral, parenteral, or neuraxial route were converted to intravenous morphine equivalent (ME) dose using a standardized dose calculator(http://www.uptodate.com/contents/cancerpainmanagement-with-opioids optimizing-analgesia). Mean and median ME per case were calculated for each calendar month from July 2017 to May 2019.

All PACU patient-reported pain score (11-point numerical score) recorded by the nursing team were averaged per case. Mean and median pain scores for each calendar month from July, 2017 to May 2019 were calculated.

## Sample size

This was a convenience sample of all eligible cases that could be included during the study period from July, 2017 to May, 2019.

## Statistical analysis

For continuous variables, descriptive statistics of cases are presented as mean, standard deviation, minimum, and maximum. For categorical variables, the number of cases in each category are presented. Chi-square tests were used to compare categorical variables (gender, ASA) and linear regression models were used to compare continuous variables (age, height, weight, BMI, case length). All tests were 2-sided and P value < 0.05 was considered to be statistically significant.

Segmented regression analyses of interrupted time series were performed examining the change in morphine equivalent per case (total, parenteral and enteral), PACU pain scores and number of non-opioid analgesic medications used before and after the intervention in April, 2018 was performed. The segmented regression analysis accounts for changes in level (for an abrupt intervention effect) and trend (increase or decrease in the slope) that follow an intervention and estimates the levels and trends separately for pre- and postintervention segments. [5] Descriptive statistics and regression models were performed in R (version 3.3.2). The segmented time-series regression analyses were performed in SPSS (version 24).

## Results

A total of 56, 451 cases were identified during the study period. After exclusions 29, 621 patients were included in the final analysis (Fig 1). Demographic data including sex, height and weight and PACU length of stay were not significantly different between the pre and post-intervention group (Table 1). Age was significantly different in the pre and post intervention period however the point estimate difference was only 0.61 years. Although there was a statistically significant difference in case length between the pre and post-intervention period (Pre intervention: 146 ± 108 minutes, Post intervention: 141 ± 113 minutes, p < 0.001) the mean

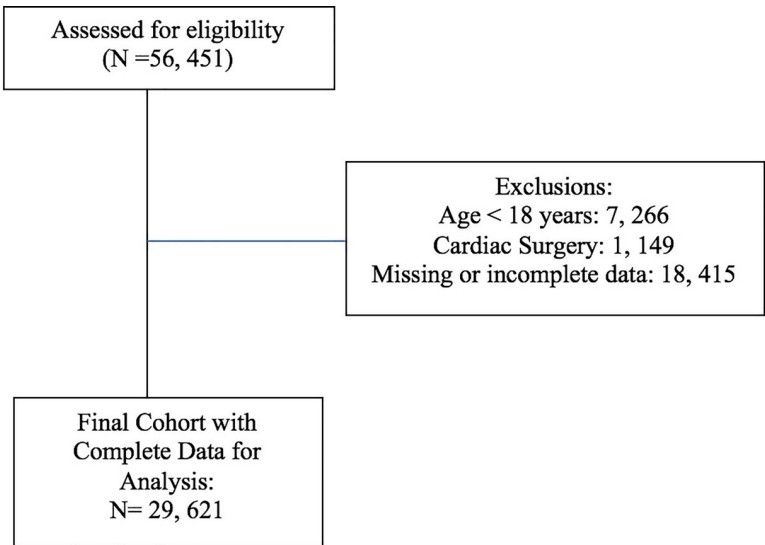

**Fig 1. Flow diagram of study participant selection.**

**Table 1. Demographic, ASA physical status and case length in the pre and post-intervention group.**

| Variable | Pre-Intervention | Post-Intervention | Estimate or $(X^2)$ | Standard Error or $(df)$ | $t$ | $p$ |
|---|---|---|---|---|---|---|
| Age | 57.2 ± 16.1 (median = 59, range = 18–100) | 57.8 ± 16.0 (median = 60, range = 18–97) | -0.61 | 0.19 | -3.20 | <0.01 |
| Height (in) | 66.9 ± 4.2 (median = 67, range = 32–82) | 67.0 ± 4.2 (median = 67, range = 43–84) | -0.02 | 0.05 | -0.34 | 0.74 |
| Gender (Female) (%) | 52.8 | 52.3 | (0.85) | (1) | | 0.36 |
| Weight (kg) | 87.1 ± 23.4 (median = 84.1, range = 40–199.4) | 87.4 ± 23.4 (median = 84.4, 40–200) | -0.31 | 0.29 | -1.06 | 0.29 |
| BMI | 30.1 ± 7.6 (median = 28.9, range = 13.7–75.5) | 30.3 ± 7.7 (median = 29.1, range = 13.8–75.9) | -0.15 | 0.10 | -1.50 | 0.13 |
| Case Length | 145.8 ± 108.5 (median = 117, range = 10–715) | 141.3 ± 108.5 (median = 111, range = 10–717) | 4.49 | 1.33 | 3.39 | < 0.001 |
| ASA Physical Status (%) | | | | | | |
| 1 | 4.7 | 4.3 | (5.9) | (5) | | 0.32 |
| 2 | 39.8 | 39.8 | | | | |
| 3 | 45.7 | 45.4 | | | | |
| 4 | 8.9 | 9.6 | | | | |
| 5 | 0.2 | 0.2 | | | | |
| 6 | 0.04 | 0.02 | | | | |
| Missing | 0.6 | 0.8 | | | | |

Chi-square tests were used to compare categorical variables (gender, ASA) and linear regression models were used to compare continuous variables (age, height, weight, BMI, case length). *df*: degrees of freedom; ASA: American Society of Anesthesiologist; BMI: Body Mass Index. Data presented as mean ± standard deviation, (mean and range) or percentage.

point difference was only 4.5 minutes. ASA physical status (Table 1, S1 Fig) and surgical procedure type (Fig 2) were similar in the pre and post-intervention period.

Trends in opioids used per case, number of nonopioids per case and patient reported pain scores before and after the intervention are reported in S1 Table and S3–S5 Figs. The segmented regression analysis of interrupted time series examining the change in morphine equivalent per case and PACU pain scores before and after the intervention in April 2018 are shown in Table 2, Fig 3 and Table 3, Fig 4 respectively. There was a significant decrease in total (Estimate: -39.9 mg, SE: 6.9 mg, p < 0.001) and parenteral opioid use (Estimate: -51.6 mg, SE: 7.1 mg, p < 0.001) after the implementation of the guideline with a significant increase in oral opioid use (Estimate: 9.4 mg, SE: 1.1 mg, p < 0.001). Furthermore, there is a sustained decrease in total (Estimate: -0.79 mg, SE 0.48 mg, p = 0.11), oral (Estimate: -0.41 mg, SE 0.11 mg, p = 0.001) and parenteral (Estimate: -0.24 mg, SE 0.49 mg, p = 0.64) opioid use following the

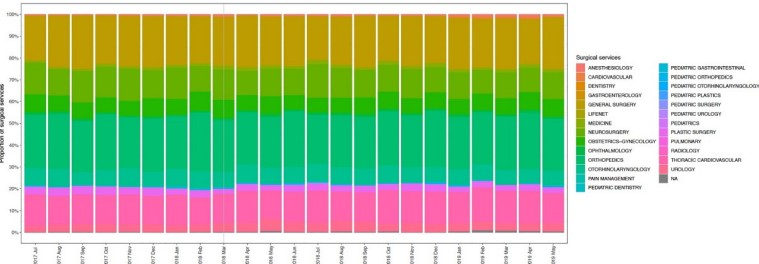

**Fig 2. Proportion of surgical procedures over time.**

**Table 2. Full segmented regression models estimating average of morphine equivalent (mg) per case before and after intervention in April, 2018.**

| | Estimate | SE | t | p |
|---|---|---|---|---|
| **Total** | | | | |
| Intercept (b0) | 86.5 | 5.4 | 16 | < 0.001 |
| Slope before April, 2018 (b1) | 1.3 | 0.97 | 1.3 | 0.21 |
| Slope change after April, 2018 (b2) | -0.8 | 0.48 | -1.6 | 0.12 |
| Level change after April, 2018 (b3) | -39.98 | 7.0 | -5.7 | < 0.001 |
| **Oral** | | | | |
| Intercept (b0) | 1.68 | 1.1 | 1.56 | 0.14 |
| Slope before April, 2018 (b1) | 0.12 | 0.18 | 0.69 | 0.5 |
| Slope change after April, 2018 (b2) | -0.41 | 0.11 | -3.87 | <0.01 |
| Level change after April, 2018 (b3) | 9.4 | 1.1 | 8.9 | < 0.001 |
| **Parenteral** | | | | |
| Intercept (b0) | 84.7 | 5.5 | 15.3 | < 0.001 |
| Slope before April, 2018 (b1) | 1.26 | 0.99 | 1.26 | 0.22 |
| Slope change after April, 2018 (b2) | -0.24 | 0.49 | -0.48 | 0.64 |
| Level change after April, 2018 (b3) | -51.56 | 7.1 | -7.25 | < 0.001 |

The intercept (b0) refers to the level of morphine equivalent use in the first month of the data series (July 2017). b1 refers to the slope of change in morphine equivalent use before April 2018. b2 refers to the difference between slope of change in morphine equivalent use before and after April, 2018, and b3 refers to the change in level of morphine equivalent use after April, 2018. SE: Standard error.

intervention. Pain scores were not significantly different between the pre and post-intervention period (Estimate: 0.05, SE: 0.13, p = 0.69).

The number of nonopioid analgesic medications utilized increased significantly after the intervention (Estimate: 0.9, SE: 0.1, p < 0.001) (Table 4, Fig 5). Trends and proportions (% per month and % change from previous month) for individual nonopioids (acetaminophen, celecoxib, dexmedetomidine, esmolol, gabapentin, ketamine and intravenous lidocaine) are represented in S2 Table and S5 Fig.

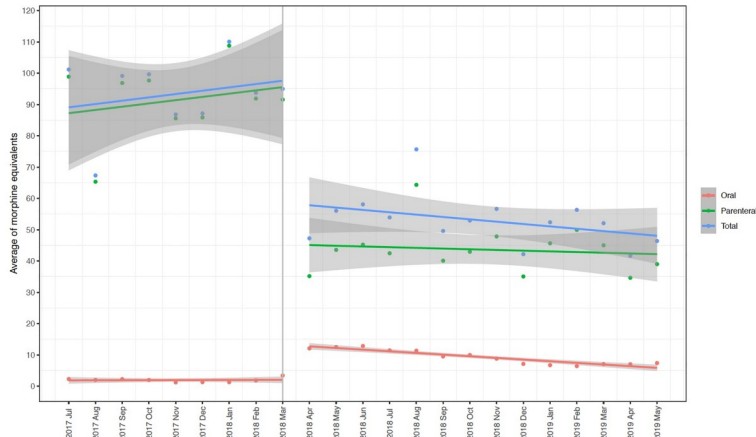

**Fig 3. Interrupted time series with segmented regression analyses demonstrating mean (standard error) total, parenteral and oral morphine equivalents per case by month before and after the intervention.**

**Table 3. Full segmented regression models estimating mean pain scores per case before and after intervention in April, 2018.**

| | Estimate | SE | t | p |
|---|---|---|---|---|
| **Total** | | | | |
| Intercept (b0) | 3.79 | 0.11 | 35.5 | < 0.001 |
| Slope before April, 2018 (b1) | -0.01 | 0.02 | -0.7 | 0.5 |
| Slope change after April, 2018 (b2) | 0.01 | 0.01 | 0.6 | 0.6 |
| Level change after April, 2018 (b3) | 0.05 | 0.13 | 0.41 | 0.69 |

The intercept (b0) refers to the level of morphine equivalent use in the first month of the data series (July 2017). b1 refers to the slope of change in morphine equivalent use before April 2018. b2 refers to the difference between slope of change in morphine equivalent use before and after April, 2018, and b3 refers to the change in level of morphine equivalent use after April, 2018. SE: Standard error.

Results from a 2-proportions z-test showed no statistically significant difference in the percent of neuraxial or loco-regional procedures performed between the pre (45.4%) and post intervention period (46.6%) [chi$^2$(1) = 3.84, 0.02, p = 0.05]. The proportion of different neuraxial and loco-regional procedure performed over the study period are represented by the mosaic plot in S6 Fig and S3 Table. No significant changes, except a reduction in the proportion of both brachial plexus [Pre: 27 ± 1.9% vs. Post: 25 ± 2.7%, p = 0.04] and epidural [Pre: 3.5 ± 0.5% vs. Post: 2.9 ± 0.6%, p = 0.02] and an increase in femoral blocks [Pre: 4.4 ± 1.9% vs. Post: 10.0 ± 1.3%, p < 0.001] were noted during the study period (S1 Table).

## Discussion

Our study demonstrates that our Limited Intravenous Opioid Guideline for perioperative pain management is associated with a decrease in total opioid use with equivalent patient reported pain scores in the PACU. Surprisingly, these findings are evident without a significant increase in the number of loco-regional and neuraxial procedures performed in the post-intervention period. This study provides a framework and protocol to manage patients during the pre, intra and immediate postoperative period when supplies of intravenous opioids are limited.

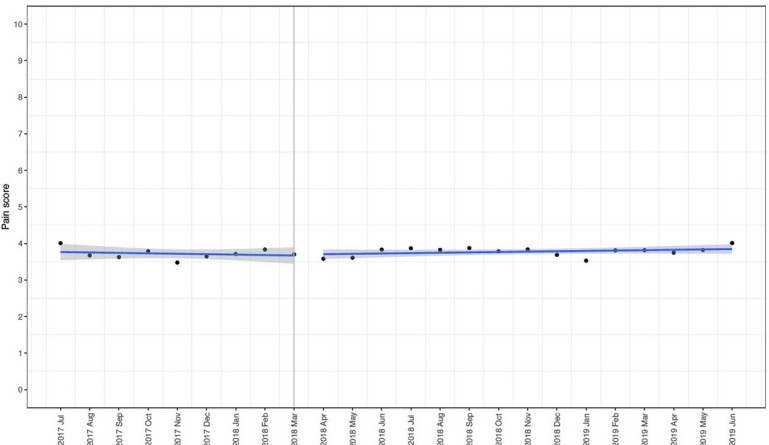

**Fig 4. Interrupted time series with segmented regression analyses demonstrating mean (standard error) post anesthesia care unit patient reported pain scores per case by month before and after the intervention.**

**Table 4. Full segmented regression models estimating mean number of non-opioid analgesics per case before and after intervention in April, 2018.**

|  | Estimate | SE | t | p |
|---|---|---|---|---|
| **Total** |  |  |  |  |
| Intercept (b0) | 2.29 | 0.07 | 35.0 | < 0.001 |
| Slope before April, 2018 (b1) | 0.01 | 0.01 | 1.4 | 0.18 |
| Slope change after April, 2018 (b2) | -0.01 | 0.01 | -1.45 | 0.16 |
| Level change after April, 2018 (b3) | 0.9 | 0.1 | 10.9 | < 0.001 |

The intercept (b0) refers to the level of morphine equivalent use in the first month of the data series (July 2017). b1 refers to the slope of change in morphine equivalent use before April 2018. b2 refers to the difference between slope of change in morphine equivalent use before and after April, 2018, and b3 refers to the change in level of morphine equivalent use after April, 2018. SE: Standard error.

Due to the current opioid epidemic, significant effort has been placed on tightening government regulation on opioid manufacturing by the Drug Enforcement Administration. Concurrently multiple manufacturing violations discovered by the FDA have resulted in significant shortages of intravenous opioids. The unintended consequences of this shortage are the risk of providing sub-standard medical care to patients undergoing surgery and those being treated for chronic cancer pain. There are several ways to mitigate this unpredictable and unreliable intravenous opioid supply chain. In an editorial in the New England Journal of Medicine, Eduardo Bruera suggested some possible solutions including strategic hospital opioid stockpiling and hospital-based compounding of common opioids. [1] However, these solutions would be unfeasible within the current federal and state regulatory structure.

An alternate solution, especially for patients being managed in the perioperative period is to optimize nonopioid multi-modal therapy and judicious use of neuraxial and loco-regional techniques. Multiple studies have demonstrated the opioid-sparing effects of acetaminophen and non-steroidal anti-inflammatory agents. [6–9] Although the gabapentenoids have reported opioid sparing effects, recent meta-analysis have failed to demonstrate beneficial effect in the perioperative period. [7, 10] However, due to the limited availability of other non-opioid analgesic agents we elected to utilize gabapentenoids for this protocol. Finally, continuous intraoperative use of lidocaine and ketamine infusions has been shown to reduce postoperative pain and opioid use in a variety of surgical procedures. [11, 12] By codifying nonopioid

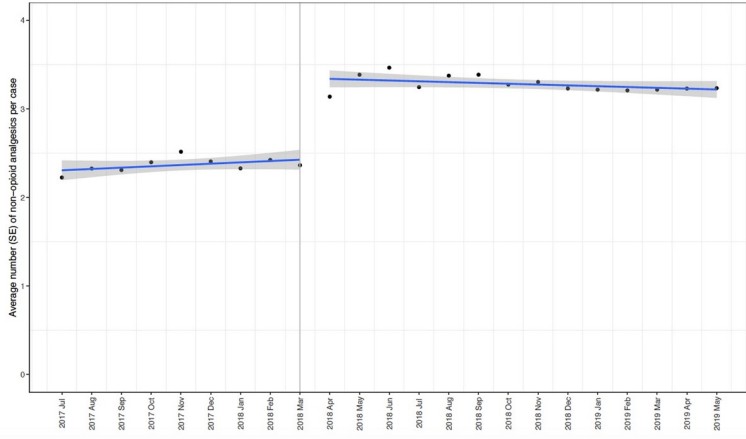

**Fig 5. Interrupted time series with segmented regression analyses demonstrating mean (standard error) number of nonopioid analgesics utilized per case by month before and after the intervention.**

analgesic options and transitioning from parenteral to predominant enteral opioid therapy, we have provided a framework to manage patients undergoing a variety of surgical procedures during periods of limited intravenous opioid availability.

An interesting finding of our study was the significant and sustained decrease in total, intravenous and oral opioid requirements after implementation of the protocol. This effect continued to be evident despite the increasingly availability (February, 2019) of intravenous opioids over the post-intervention period. These findings are similar to the results reported by Ackerman et al. with their experience on an intravenous to oral and subcutaneous substitution policy in a general medical unit. [4] They demonstrated that intravenous opioid doses were reduced by 84% (0.06 vs 0.39 doses per patient-day, P < .001), doses of all parenteral opioids were reduced by 55% (0.18 vs 0.39 doses per patient-day, P < .001) while mean daily overall opioid exposure decreased by 31% (6.30 [4.12] vs 9.11 [7.34] MMEs per patient-day). [4] Of note, for hospital days 1 through 3, there were no significant postintervention vs. preintervention differences in mean reported pain score for patients receiving opioid therapy: day 1, -0.19 (95% CI, -0.94 to 0.56); day 2, -0.49 (95% CI, -1.01 to 0.03); and day 3, -0.54 (95% CI, -1.18 to 0.09).This finding can be partially explained by the increase use of nonopioid adjuncts in the post-intervention period. Furthermore, there is compelling evidence for the role of intraoperative opioid-induced hyperalgesia increasing postoperative opioid requirements. [13] Limiting the dose of intravenous opioids administered potentially reduces the risk of opioid-induced hyperalgesia and therefore patients require less opioids in the postoperative period. However additional, prospective studies are needed to better elucidate this effect. The impact of a sustained prescribing behavioral change amongst our anesthesia providers cannot be under-estimated. Sustainable behavior in the healthcare setting is successful when barriers are limited and when cogent policy changes are mandated, as exemplified by our pragmatic intervention. [14, 15] Finally, our protocol provides a practical framework for perioperative pain management in low and middle income countries where the supply of intravenous opioids is limited or unreliable. [16, 17]

Our study has a significant limitation: the primary aim of this study was to provide a framework to manage patients during the pre, intra and immediate postoperative period, we therefore did not evaluate the impact of our intervention on opioid use and patient reported pain scores after discharge from the PACU. Additional studies are needed to evaluate whether increased use of multi-modal analgesia and a predominant enterally-based opioid analgesic regimen in the immediate perioperative period is associated with reduced hospital and post-discharge opioid requirements. Secondly, dexamethasone was not included as a nonopioid adjunct due to its major benefits confined to pediatric patients undergoing tonsillectomy and as an adjunct to peripheral nerve block procedures. Finally, preoperative pain scores and opioid use were not reliably documented in our electronic medical record and were not included in this study. The impact of the aforementioned factors on postoperative pain and analgesic requirements are well known and is a limitation of this study.

## Conclusion

We demonstrate that an enteral based opioid regimen for perioperative analgesia management, anchored by multi-modal nonopioid adjuncts and neuraxial/loco-regional techniques, can mitigate the impact of an intravenous opioid shortage.

## Supporting information

**S1 Checklist. STROBE statement—checklist of items that should be included in reports of observational studies.**
(DOCX)

**S1 Fig. Mosaic plot of ASA physical status proportion in the pre and post-intervention period.** NA: Cases with missing ASA classification.
(TIFF)

**S2 Fig. Trend graph of total, oral and parenteral average morphine equivalents in the pre and post-intervention period.**
(TIFF)

**S3 Fig. Trend graph of the number of nonopioid analgesics per case in the pre and post-intervention period.**
(TIFF)

**S4 Fig. Trend graph of average pain score in the pre and post-intervention period.**
(TIFF)

**S5 Fig. Trend graph of the proportion of specific nonopioid analgesic agents in the pre and post-intervention period.**
(TIFF)

**S6 Fig. Mosaic plot of loco-regional and neuraxial distribution frequency in the pre and post-intervention period.**
(TIFF)

**S1 Table. Mean (standard deviation), median[interquartile range] and minimum and maximum total, oral and parenteral morphine equivalents in the pre and post intervention period.** Pre-pre intervention, post-post intervention.
(DOCX)

**S2 Table. Changes in the percentage of nonopioids administered pre and post intervention.** Data presented as percentage and percentage change from previous month. Pre-pre intervention, post-post intervention.
(DOCX)

**S3 Table. Changes in the proportion of block groups pre and post intervention.** TAP: Transverse abdominis plane.
(DOCX)

**S1 Dataset.**
(XLSX)

**S2 Dataset.**
(XLSX)

## Acknowledgments

We thank Amir Abdel-Malik for his assistance in data extraction and data management for this study.

## Author Contributions

**Conceptualization:** Reza Salajegheh, Edward C. Nemergut, Bhiken I. Naik.

**Data curation:** Reza Salajegheh, Terran M. Rice, Roy Joseph, Siny Tsang, Bethany M. Sarosiek, C. Paige Muthusubramanian, Katelyn M. Hipwell, Kate B. Horton.

**Formal analysis:** Siny Tsang, Bhiken I. Naik.

**Investigation:** Bhiken I. Naik.

**Methodology:** Bhiken I. Naik.

**Supervision:** Bhiken I. Naik.

**Writing – original draft:** Reza Salajegheh, Edward C. Nemergut, Bhiken I. Naik.

**Writing – review & editing:** Reza Salajegheh, Edward C. Nemergut, Terran M. Rice, Roy Joseph, Siny Tsang, Bethany M. Sarosiek, C. Paige Muthusubramanian, Katelyn M. Hipwell, Kate B. Horton, Bhiken I. Naik.

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
