## [Decision Letter · Decision Letter 0]

9 Apr 2020

PONE-D-20-02238

Impact of A Perioperative Oral Opioid Substitution Protocol During The Nationwide Intravenous Opioid Shortage: A Single Center, Interrupted Time Series with Segmented Regression Analysis

PLOS ONE

Dear Dr. Naik,

Thank you for submitting your manuscript to PLOS ONE. After careful consideration, we feel that it has merit but does not fully meet PLOS ONE’s publication criteria as it currently stands. Therefore, we invite you to submit a revised version of the manuscript that addresses the points raised during the review process.

We would appreciate receiving your revised manuscript by May 24 2020 11:59PM. To enhance the reproducibility of your results, we recommend that if applicable you deposit your laboratory protocols in protocols.io, where a protocol can be assigned its own identifier (DOI) such that it can be cited independently in the future. For instructions see: http://journals.plos.org/plosone/s/submission-guidelines#loc-laboratory-protocols

We look forward to receiving your revised manuscript.

Kind regards,

Patrice Forget

Academic Editor

PLOS ONE

2. In the ethics statement in the manuscript and in the online submission form, please provide additional information about the patient records used in your retrospective study, including: a) whether all data were fully anonymized before you accessed them; b) the date range (month and year) during which patients' medical records were accessed; and c) the source of the medical records analyzed in this work (e.g. hospital, institution or medical center name). If patients provided informed written consent to have data from their medical records used in research, please include this information.

Reviewers' comments:

Reviewer's Responses to Questions

**Comments to the Author**

1. Is the manuscript technically sound, and do the data support the conclusions?

Reviewer #1: No

Reviewer #2: Yes

2. Has the statistical analysis been performed appropriately and rigorously? 

Reviewer #1: No

Reviewer #2: I Don't Know

3. Have the authors made all data underlying the findings in their manuscript fully available?

Reviewer #1: No

Reviewer #2: Yes

4. Is the manuscript presented in an intelligible fashion and written in standard English?

Reviewer #1: No

Reviewer #2: Yes

5. Review Comments to the Author

Reviewer #1: The authors report an interesting large cohort study. Due to an unanticipated shortage of intravenous opioids, the authors had to follow the multimodal non-opioid analgesia procedures widely described in the literature. Their results are in line with all the literature. They confirm the potential hyperalgesia induced by opioids.

If the segmented regression analysis is interesting, it remains difficult to follow to be didactic. This aspect needs to be improved.

+ Give results not only with mean (SE) but also with SD and with median (IQR), Max and Min values.

+ Has dexamethasone been used in some cases?

+ Fig 1: summary of ASA classification as not significant; but put the proportions of the different surgeries (not as supplemented data). Is it possible to have the level of preoperative pain, if there is opioid tolerant patient and so (beta-blockers …)? Differentiate the presentation in tables with a "classic" presentation and the figs in trend curves.

+ What is the possible explanation for a reduced length of time (duration of surgery Vs. duration of anesthesia timing?). A correlation graph between the amount of morphine used during the anesthesia and postoperatively would be useful. A presentation of the potential correlation between postoperative pain and postoperative consumption of morphine would also be useful (in PACU and on the floor). In the same line, can the side effects related to opioids (such as PONV or at least their treatment) and to the molecules used in substitution (such as bradycardia, sedation) be reported? Was there an effect on the length of hospital stay for the same surgical procedure?

+ Because the intraoperative paradigm shift (opioid-based anesthesia to opioid-reduced anesthesia or opioid-free anesthesia) which reactivity (i.e. pain) monitoring was used during the anesthesia?

Reviewer #2: In my opinion, an analysis of the non-opioid agents that are used to make morphine sparing is missing. Indeed, the message in the present state is that morphine sparing is mainly a cultural problem and that by using more non-morphine agents the immediate perioperative consumption of morphine is reduced. There is not enough development on the molecules that need to be used more (dexmedetomidine according to your data?). The dose of the drug is not raised, but it is likely that some (such as ketamine) will be used at higher doses, which I think should be said in the discussion.

6. PLOS authors have the option to publish the peer review history of their article (what does this mean?). If published, this will include your full peer review and any attached files.

Reviewer #1: No

Reviewer #2: Yes: Dr Vincent COLLANGE, MD of anesthesiology, Medipole Lyon Villeurbanne, France.

---

## [Author Response · Author response to Decision Letter 0]

1 May 2020

Response to reviewers 

Reviewer #1: The authors report an interesting large cohort study. Due to an unanticipated shortage of intravenous opioids, the authors had to follow the multimodal non-opioid analgesia procedures widely described in the literature. Their results are in line with all the literature. They confirm the potential hyperalgesia induced by opioids.

If the segmented regression analysis is interesting, it remains difficult to follow to be didactic. This aspect needs to be improved.

+ Give results not only with mean (SE) but also with SD and with median (IQR), Max and 

In the revised manuscript, we included median, minimum, and maximum values in the descriptive statistics (Table 1). We have also provided an additional table (Supplementary Table 1) that reports the mean (standard deviation), median[interquartile range] and minimum-maximum for total, parenteral and oral opioids administered as requested by the reviewer. Results of the interrupted time series analyses are presented as means and standard errors (SEs). The SEs reflect the precision of the estimates (i.e., the means). We disagree with this reviewer’s suggestion to include standard deviations (SDs), as SDs reflect only the variability of a variable, not the precision of the estimated mean. 

+ Has dexamethasone been used in some cases?

Dexamethasone was not included in this analysis. However, we are aware of several studies that have documented the small but statistically significant reduction in opioid use with dexamethasone. We have included this as a limitation in the ‘Discussion’. 

+ Fig 1: summary of ASA classification as not significant; but put the proportions of the different surgeries (not as supplemented data). 

We have added the proportions of different surgical services in the manuscript (Fig 2: Proportion of surgical procedures over time)

Is it possible to have the level of preoperative pain, if there is opioid tolerant patient and so (beta-blockers …)? 

Thank you for this insightful comment. Preoperative pain scores were not reliably collected before surgery therefore we did not include them in the analysis. We have added this to the limitations in the ‘Discussion’ (Page 21, line 354-357).

Differentiate the presentation in tables with a "classic" presentation and the figs in trend curves.

In the revised manuscript, we included descriptive statistics by month and additional figures in the supplementary material to illustrate the trends in the data (Supplementary Fig 2-5). 

+ What is the possible explanation for a reduced length of time (duration of surgery Vs. duration of anesthesia timing?). 

It’s important to differentiate a ‘statistically’ significant result in such a large dataset vs. the clinically meaningful effect size. The effect size of case length here is 4.5 (1.33) minutes, which is not clinically meaningful in our opinion. This is elaborated in the first paragraph of the Results section.

A correlation graph between the amount of morphine used during the anesthesia and postoperatively would be useful. 

We did not parse out the cumulative morphine requirements by pre/intraoperative and post anesthesia care unit. This is primarily due to difficulty in anchoring dose administration for these different time periods due to how they are recorded in the electronic medical record. We therefore did not perform this analysis however we recognize the importance of this analysis. 

A presentation of the potential correlation between postoperative pain and postoperative consumption of morphine would also be useful (in PACU and on the floor). 

We did not collect any medication data or pain scores from the wards. 

In the same line, can the side effects related to opioids (such as PONV or at least their treatment) and to the molecules used in substitution (such as bradycardia, sedation) be reported? Was there an effect on the length of hospital stay for the same surgical procedure?

We did not collect data for complications in this study. Furthermore, the study was confined to the intraoperative and immediate postoperative period only. The primary goal of this study was to present a framework for the use of oral opioids supplemented by nonopioid analgesics in the event of reduced intravenous opioid availability (example during the current COVID pandemic)

+ Because the intraoperative paradigm shift (opioid-based anesthesia to opioid-reduced anesthesia or opioid-free anesthesia) which reactivity (i.e. pain) monitoring was used during the anesthesia?

The indication for administering intraoperative intravenous opioids are described in the section: Intraoperative Phase of Care (Point 8) on page 9, ln 156-157 and ln 161-162. Primarily hypertension and tachycardia, after excluding inadequate depth of anesthesia, was the trigger to administer intravenous opioids. 

Reviewer #2: In my opinion, an analysis of the non-opioid agents that are used to make morphine sparing is missing. 

Thank you for this excellent suggestion. We have provided both a table (Supplementary Table 2) and trend graph of nonopioid analgesics (Supplementary Figure 5) administered during the pre and post-intervention. This provides the reader with information on how the frequency of nonopioid analgesic use changed during the pre and post intervention period. 

Indeed, the message in the present state is that morphine sparing is mainly a cultural problem and that by using more non-morphine agents the immediate perioperative consumption of morphine is reduced. 

We humbly disagree with the reviewer on this point. We feel that there are two factors at play: increased nonopioid use (on average one additional nonopioid use) and the phenomena of reduced opioid induced hyperalgesia with less total intraoperative opioid used, as highlighted by reviewer 1. 

There is not enough development on the molecules that need to be used more (dexmedetomidine according to your data?). The dose of the drug is not raised, but it is likely that some (such as ketamine) will be used at higher doses, which I think should be said in the discussion.

We have provided an ITS on the number of nonopioid analgesics administered during the case. Per the reviewer’s earlier suggestion, we have provided both a table and trend graph of nonopioid analgesics administered during the pre and post-intervention. This provides the reader with data on how the frequency of nonopioid analgesic use changed during the pre and post intervention period. Doses of medications were not changed during the post intervention period.

---

## [Decision Letter · Decision Letter 1]

19 May 2020

PONE-D-20-02238R1

Impact of A Perioperative Oral Opioid Substitution Protocol During The Nationwide Intravenous Opioid Shortage: A Single Center, Interrupted Time Series with Segmented Regression Analysis

PLOS ONE

Dear Dr. Naik,

Thank you for submitting your manuscript to PLOS ONE. After careful consideration, we feel that it has merit but does not fully meet PLOS ONE’s publication criteria as it currently stands. Therefore, we invite you to submit a revised version of the manuscript that addresses the last point raised during the review process.

Please consider the reviewer's comment and amend your discussion accordingly: "The modifications with the non-opioid molecules may have been introduced before the change in practice or without any change. This could only partially be due to the change in the route of administration of opioids."

Please consider the "minor points" too.

We would appreciate receiving your revised manuscript by Jul 03 2020 11:59PM. To enhance the reproducibility of your results, we recommend that if applicable you deposit your laboratory protocols in protocols.io, where a protocol can be assigned its own identifier (DOI) such that it can be cited independently in the future. For instructions see: http://journals.plos.org/plosone/s/submission-guidelines#loc-laboratory-protocols

A rebuttal letter that responds to the point raised by the academic editor and reviewer(s). This letter should be uploaded as separate file and labeled 'Response to Reviewers'.A marked-up copy of your manuscript that highlights changes made to the original version. This file should be uploaded as separate file and labeled 'Revised Manuscript with Track Changes'.An unmarked version of your revised paper without tracked changes. This file should be uploaded as separate file and labeled 'Manuscript'.

We look forward to receiving your revised manuscript.

Kind regards,

Patrice Forget

Academic Editor

PLOS ONE

Reviewers' comments:

Reviewer's Responses to Questions

**Comments to the Author**

1. If the authors have adequately addressed your comments raised in a previous round of review and you feel that this manuscript is now acceptable for publication, you may indicate that here to bypass the “Comments to the Author” section, enter your conflict of interest statement in the “Confidential to Editor” section, and submit your "Accept" recommendation.

Reviewer #1: All comments have been addressed

Reviewer #2: All comments have been addressed

2. Is the manuscript technically sound, and do the data support the conclusions?

Reviewer #1: Partly

Reviewer #2: Yes

3. Has the statistical analysis been performed appropriately and rigorously? 

Reviewer #1: I Don't Know

Reviewer #2: Yes

4. Have the authors made all data underlying the findings in their manuscript fully available?

Reviewer #1: Yes

Reviewer #2: Yes

5. Is the manuscript presented in an intelligible fashion and written in standard English?

Reviewer #1: Yes

Reviewer #2: Yes

6. Review Comments to the Author

Reviewer #1: I carefully reread the revised version of the manuscript, the questions and the answers. I am perplexed about the didactic interest of this work. Indeed, if the evolution was made due to a restriction of the parenteral opioid stock, it also corresponds to the classic evolution of practices with a more extensive use of non-opioid analgesic drugs. This current work confirms these results: more non-opioid used allows less opioid for, at least, the same effectiveness. If we look carefully at the modifications with the non-opioid molecules; it seems to have been introduced before the change in practice (see: dexmedetomidine, acetaminophen and celecoxib) or without any change (lidocaine, esmolol). Would this be due only to the change in the route of administration of opioids (i.e. oral Vs. parenteral)? I’m not so sure.

Minor points.

+ The use of gabapentin preoperatively is less and less recommended on the basis of meta-analysis.

+ P7 L1119-120 “prior to surgery, add during the procedure, and in the PACU”

+ Can't the fact that the patients were significantly older, with significantly shorter procedures, be responsible for these results?

Reviewer #2: (No Response)

7. PLOS authors have the option to publish the peer review history of their article (what does this mean?). If published, this will include your full peer review and any attached files.

Reviewer #1: No

Reviewer #2: Yes: Vincent Collange MD, Anesthesiology, Medipole Lyon Villeurbanne

---

## [Author Response · Author response to Decision Letter 1]

20 May 2020

Reviewer #1:

 I carefully reread the revised version of the manuscript, the questions and the answers. I am perplexed about the didactic interest of this work. Indeed, if the evolution was made due to a restriction of the parenteral opioid stock, it also corresponds to the classic evolution of practices with a more extensive use of non-opioid analgesic drugs. This current work confirms these results: more non-opioid used allows less opioid for, at least, the same effectiveness.

Response: We agree with the reviewer that increased nonopioid analgesic use (based on a protocol) definitely has an opioid sparing effect. This study confirms this well described effect. However, the study also elucidates the comparative benefits of an oral opioid regimen that can be utilized when parenteral opioids are limited. Furthermore, our study also tentatively explores the effect of opioid induced hyperalgesia caused by high dose parenteral opioids, which may be mitigated by oral opioids. This is reported extensively in the ‘Discussion’ section. 

If we look carefully at the modifications with the non-opioid molecules; it seems to have been introduced before the change in practice (see: dexmedetomidine, acetaminophen and celecoxib) or without any change (lidocaine, esmolol). 

Response: The changes in nonopoid use varied slightly between agents but Supplementary table 2 clears demonstrates the % changes when the protocol was formally started in April, 2018 compared to the previous month[Acetaminophen (54 to 72%), celecoxib (13 to 18-28%), dexmedetomidine (31 to 45%). 

Would this be due only to the change in the route of administration of opioids (i.e. oral Vs. parenteral)? I’m not so sure.

Response: Our study tentatively explores the effect of opioid induced hyperalgesia caused by high dose parenteral opioids, which may be mitigated by oral opioids. We discuss the findings of the study by Ackerman et al who demonstrated reduced opioid use when they transitioned from a parenteral to oral opioid regimen (Page 20, line 331-345). This is reported extensively in the ‘Discussion’ section.

Minor points.

+ The use of gabapentin preoperatively is less and less recommended on the basis of meta-analysis.

Response: We agree with the reviewer and have included this in our Discussion (Page 20, line 318-321)

+ P7 L1119-120 “prior to surgery, add during the procedure, and in the PACU”

Response: We have added this to the manuscript. Page 7, line 120-121

+ Can't the fact that the patients were significantly older, with significantly shorter procedures, be responsible for these results?

Response: We humbly disagree with the reviewer for the following reasons: 1) the cohort size for this study is large (n=29, 621), therefore small changes may be statistically significant. However, it is important to interpret the effect size of the change in age and procedure length, which is included in Table 1. The effect size for age is 0.61 years, while the procedure length is 4.49 minutes. These changes are not clinically relevant. The following reference is critical to understanding our response : Using Effect Size—or Why the P Value Is Not Enough. 

J Grad Med Educ. 2012 Sep; 4(3): 279–282. PMID: 23997866. We have addressed this on page 12, line 217-221.

---

## [Editor Report · Decision Letter 2]

21 May 2020

Impact of A Perioperative Oral Opioid Substitution Protocol During The Nationwide Intravenous Opioid Shortage: A Single Center, Interrupted Time Series with Segmented Regression Analysis

PONE-D-20-02238R2

Dear Dr. Naik,

We are pleased to inform you that your manuscript has been judged scientifically suitable for publication and will be formally accepted for publication once it complies with all outstanding technical requirements.

With kind regards,

Patrice Forget

Academic Editor

PLOS ONE
---

## [Editor Report · Acceptance letter]

26 May 2020

PONE-D-20-02238R2 

Impact of A Perioperative Oral Opioid Substitution Protocol During The Nationwide Intravenous Opioid Shortage: A Single Center, Interrupted Time Series with Segmented Regression Analysis 

Dear Dr. Naik:

I am pleased to inform you that your manuscript has been deemed suitable for publication in PLOS ONE. Congratulations! Your manuscript is now with our production department. 

With kind regards,

on behalf of

Prof. Patrice Forget 

Academic Editor

PLOS ONE